Transcriptome and structure analysis in root of Casuarina equisetifolia under NaCl treatment

Wang Yujiao 1
Zhang Jin 2
Qiu Zhenfei 1
Zeng Bingshan 1
Zhang Yong 1
Wang Xiaoping 1
Chen Jun 1
Zhong Chonglu 1
Deng Rufang 3
Fan Chunjie fanchunjie@caf.ac.cn 1
1 State Key Laboratory of Tree Genetics and Breeding, Key Laboratory of State Forestry and Grassland Administration on Tropical Forestry, Research Institute of Tropical Forestry, Chinese Academy of Forestry , Guangzhou , China
2 State Key Laboratory of Subtropical Silviculture, School of Forestry and Biotechnology, Zhejiang A&F University , Hangzhou , Zhejiang , China
3 South China Botanical Garden, Chinese Academy of Sciences , Guangzhou , China
Lazo Gerard
Electronic publication date: 2021 Sep 22
Publication date: 2021
Volume: 9
Electronic Location ID: e12133
Received 2021 Mar 5; Accepted 2021 Aug 18
Copyright: ©2021 Wang et al.
Copyright year: 2021
Copyright holder: Wang et al.
License: This is an open access article distributed under the terms of the Creative Commons Attribution License, which permits unrestricted use, distribution, reproduction and adaptation in any medium and for any purpose provided that it is properly attributed. For attribution, the original author(s), title, publication source (PeerJ) and either DOI or URL of the article must be cited.
License URL: https://creativecommons.org/licenses/by/4.0/

Keywords: Salt stress, Differentially expressed genes, Epidermal cells, Ion content, Programmed cell death

Funding: The Specific Program for National Non-profit Scientific Institutions CAFYBB2018ZB003 The National Natural Science Foundation of China 31770716 This work was supported by a grant from the Specific Program for National Non-profit Scientific Institutions (CAFYBB2018ZB003), a project funded by the National Natural Science Foundation of China (Grant No. 31770716). The funders had no role in study design, data collection and analysis, decision to publish, or preparation of the manuscript.

==============================
Background

High soil salinity seriously affects plant growth and development. Excessive salt ions mainly cause damage by inducing osmotic stress, ion toxicity, and oxidation stress. Casuarina equisetifolia is a highly salt-tolerant plant, commonly grown as wind belts in coastal areas with sandy soils. However, little is known about its physiology and the molecular mechanism of its response to salt stress.

Results

Eight-week-old C. equisetifolia seedlings grown from rooted cuttings were exposed to salt stress for varying durations (0, 1, 6, 24, and 168 h under 200 mM NaCl) and their ion contents, cellular structure, and transcriptomes were analyzed. Potassium concentration decreased slowly between 1 h and 24 h after initiation of salt treatment, while the content of potassium was significantly lower after 168 h of salt treatment. Root epidermal cells were shed and a more compact layer of cells formed as the treatment duration increased. Salt stress led to deformation of cells and damage to mitochondria in the epidermis and endodermis, whereas stele cells suffered less damage. Transcriptome analysis identified 10,378 differentially expressed genes (DEGs), with more genes showing differential expression after 24 h and 168 h of exposure than after shorter durations of exposure to salinity. Signal transduction and ion transport genes such as HKT and CHX were enriched among DEGs in the early stages (1 h or 6 h) of salt stress, while expression of genes involved in programmed cell death was significantly upregulated at 168 h, corresponding to changes in ion contents and cell structure of roots. Oxidative stress and detoxification genes were also expressed differentially and were enriched among DEGs at different stages.

Conclusions

These results not only elucidate the mechanism and the molecular pathway governing salt tolerance, but also serve as a basis for identifying gene function related to salt stress in C. equisetifolia.

Introduction

Salinity resulting mainly from sodium chloride (NaCl) is one of the most severe environmental stresses on plants. Salt stress damages root structure, hindering the absorption of nutrients by plant roots, which affects normal growth and development. The perception of Na+ in roots is a rapid process, leading to osmotic stress and a rapid increase in the cytoplasmic Ca2+ concentration of roots (Choi et al., 2014; Deinlein et al., 2014). Kinases associated with Ca2+ signaling such as calcium dependent protein kinases (CDPKs), calcineurin B-like proteins (CBLs), and protein kinases (CIPKs) further regulate downstream protein activity and gene transcription (Boudsocq & Sheen, 2013; Das & Pandey, 2010; Weinl & Kudla, 2009). Small amounts of reactive oxygen species (ROS) are used as signaling molecules in response to salt stresses; however, high concentrations cause oxidative damage (Laloi, Apel & Danon, 2004; Suzuki et al., 2012; Gill & Tuteja, 2010). After sensing salt stress signals, plants respond to stress by regulating Na+ transport and maintaining ion balance to reduce damage (Morton et al., 2019; Van Zelm, Zhang & Testerink, 2020). OsHKT1 transports Na+ into root cells in rice, compensating for K+ deficiency (Horie et al., 2001). Overexpression of StNHX1, OsHAK5, and HbHAK1 makes plants highly tolerant to salt stress (Chen et al., 2014; Zhang et al., 2020; Horie et al., 2011). The SOS (salt overly sensitive) signaling pathway is a well-studied pathway closely related to plant salt tolerance (Qiu et al., 2002). SOS2 and SOS3 act together to regulate the Na+/H+ anti-transporter SOS1 (Halfter, Ishitani & Zhu, 2000), which occurs mainly in roots and maintains intracellular and extracellular ion balance by transporting Na+ to the extracellular space, relieving ionic toxicity (Ishitani et al., 2000). Phytohormones are crucial endogenous chemical signals coordinating plant growth and environmental challenges (Park, Kim & Yun, 2016). For example, abscisic acid (ABA) and jasmonic acid (JA) play important roles in helping plants to cope with salt stress by growth repression or sensing signals (Yu et al., 2020; Chen et al., 2016; Geng et al., 2013). Furthermore, gene ontology (GO) terms for plant hormone pathways in the selected gene modules of Populus are enriched under salt stress (Liu et al., 2019). These results suggest that salt tolerance is a complex trait that involves a coordinated response to osmotic and ionic stresses and their subsequent secondary stresses.

Casuarina, an angiosperm plant widely grown in the tropics and subtropics, is the main tree genus in coastal shelter forests of south-eastern China where it acts as a coastal windbreak and stabilizes sand (Zhong et al., 2001; Zhong et al., 2010). Meanwhile, it also forms a symbiosis with the actinomycete Frankia in its natural habitat, forming root nodules and fixing nitrogen from air. Casuarina glauca and Casuarina equisetifolia are highly salt-tolerant species (Aswathappa & Bachelard, 1986). A previous study showed that C. equisetifolia seedlings can survive in 500 mM NaCl solution and even form root nodules in 300 mM NaCl solution (Tani & Sasakawa, 2003). The symbiotic system between C. equisetifolia seedlings and the Frankia Ceql strain also shows high tolerance to salt (Tani & Sasakawa, 2003). Salt tolerance of some C. equisetifolia clones is related to the rate of germination, seedling height, and proline content (Wu et al., 2010). Recently, a few studies have attempted to explain the mechanism of physiological and molecular responses to salt stress in Casaurina. Under salt stress, proline accumulation occurs to adjust the osmotic pressure; however, glycine betaine, other amino acids, and total sugars in C. equisetifolia remain unchanged (Selvakesavan et al., 2016; Tani & Sasakawa, 2010). Similarly, C. glauca tolerates high levels of salinity by changing the levels of some neutral sugars, proline, and ornithine (Jorge et al., 2017). In addition, greater amounts of Na+ are adsorbed over the roots under salt stress, and expression of NHX and SOS genes in roots helps to maintain K+ balance, an essential part of the response to excess salt in C. equisetifolia (Fan et al., 2018). However, the unique strategies adopted by these plants for dealing with salinity and the salt-tolerance mechanisms of these species remain unclear.

The emergence of sequencing technology directly and profoundly revealed the deep information of nucleic acid molecules, and provided a decisive technical means for further exploration of gene structure and function. Fortunately, the recent publication of the genome of C. equisetifolia (Ye et al., 2019) provides valuable information for studying such mechanisms. The obtained sequencing data were compared and spliced with the reference genome. Furthermore, annotation and description of transcripts based on genomic data were added. Transcriptome analysis enables us to identify differentially expressed genes (DEGs) by providing comprehensive mRNA profiles.

Sodium sensors in roots allow plants to respond quickly to stress, such as the rapid salt-specific response in roots and the rapid, sodium-specific effect of salt on root growth direction (salt solubility) (Choi et al., 2014; Galvan-Ampudia et al., 2013). Roots of C. equisetifolia are tolerant of salt stress in saline soil (Tani & Sasakawa, 2003; Fan et al., 2018). However, there are few studies on the response mechanism in C. equisetifolia roots under salt stress. The present study therefore focused on the cell structure and a comprehensive transcriptome analysis of C. equisetifolia roots exposed to salinity in the form of a 200 mM NaCl solution for varying durations. The specific objectives were (1) to obtain an overview of the changes over time in C. equisetifolia roots under salt stress to complement the insights into the molecular mechanisms of salt tolerance and alterations in root structure in plants, and (2) to identify a number of candidate genes that can be exploited in breeding for enhancing salt tolerance.

Material and methods

Plant material and salt stress

The C. equisetifolia clone A8 was preserved and cultivated by the Research Institute of Tropical Forestry, Chinese Academy of Forestry, Guangzhou. The methods used here refer to previous studies (Jiang & Deyholos, 2006; Kawasaki et al., 2001). Rooted cuttings of C. equisetifolia clone A8 cultured in a growth chamber for 8 weeks were prepared for the experiment. Their roots were washed and the plants were transferred to containers filled with clean water and allowed to grow for 2 weeks until new roots appeared. Plants were then transferred to 12 Hoagland solution containing 200 mM NaCl based on a previous study (Fan et al., 2018) and allowed to grow for varying durations. The solution was replaced with fresh solution every day. Roots were harvested following 0, 1, 6, 24, and 168 h of salt treatment (Jiang & Deyholos, 2006) and stored at −80 °C until further analysis. The experiment was arranged such that all samples were harvested at the same time. Three replicates were processed at time point, with three individuals in each replicate. Root samples were finely ground, and 1.0 g of the plant material was digested with concentrated HNO3 in a microwave digestion system (Mars 5, CEM Corporation, Matthews, North Carolina, USA). Inductively coupled plasma emission spectrometry (ICP-oes, Varian Vista-Pro RL) was used to determine the contents of Na+, K+ and Cl− in the extract. These experiments were repeated at least in triplicate, and significant differences between mean values were analyzed using one-way ANOVA: the software employed was IBM SPSS Statistics ver. 19 for Windows.

Preparation and observation of tissue morphology and ultrastructure

Newly produced roots from the same position on each seedling were selected, cut into pieces approximately 1-2 mm, and fixed in 0.1 M phosphate buffer (pH 7.2) containing 2% glutaraldehyde and 2.5% paraformaldehyde. Root samples were washed six times with 0.1 M phosphate buffer, fixed in 1% osmium tetroxide for 4 h, and washed again with 0.1 M phosphate buffer. Fixed root samples were then dehydrated and embedded in a flat mold using EPON812 resin. Semi-thin sections (1 µm) and ultrathin sections (80 nm) were cut with an ultramicrotome (Leica EM UC7). The semi-thin sections were stained with toluidine blue and the cell structure was examined under a light microscope (Olympus AX70). Ultrathin sections were stained with 4% uranyl acetate and 2% lead citrate and were examined under a transmission electron microscope (JEM-1010; JEOL, Tokyo, Japan) operating at 100 kV.

Extraction and purification of RNA

New roots were collected from each individual plant and stored in liquid nitrogen at −80 °C prior to RNA extraction. Total RNA from each sample was isolated separately using an RN38 EASYspin plus Plant RNA kit (Aidlab Biotech, Beijing, China) following the manufacturer’s instructions. Purified RNA was quantified using a NanoDrop 2000 spectrophotometer (ThermoFisher Scientific, Wilmington, Delaware, USA), and RNA integrity was evaluated using an Agilent 2100 Bioanalyzer (Agilent Technologies, Santa Clara, California, USA). Three replicates were processed at time point, with three individuals in each replicate. For each sample, at least 20 µg of total RNA was sent to Suzhou Encode Genomics Bio-technology Co., Ltd, for Illumina sequencing. Meanwhile, one copy of each RNA sample was kept in the −80 °C refrigerator for quantitative reverse-transcription polymerase chain reaction (qRT-PCR) experiments. Based on the manufacturer’s instructions, 1–2 µg of total RNA was used as a template in RT reactions with SuperScript III reverse transcriptase (Invitrogen; Thermo Fisher Scientific).

mRNA-Seq experiment and transcriptome assembly

Ten samples were collected and labelled as follows to reflect the duration of stress and replication: control -1, control -2, 1h-1, 1h-2, 6h-1, 6h-2, 24h-1, 24h-2, 168h-1, and 168h-2.

After sequencing, the raw sequence data were initially processed to obtain clean reads by removing adapter sequences and low-quality sequences. The genomic sequence of Casuarina and the corresponding GFF file were downloaded from the Casuarina SMRT database (http://forestry.fafu.edu.cn/db/Casuarinaceae/index.php). Reads of samples were aligned to the corresponding reference genome using the software package HISAT2 (Daehwan, Ben & SS, 2015) with default parameters. The SAM files were converted into BAM (binary) files and sorted with default parameters using SAMtools, and the ratio of mapped reads to reference sequences in each data set was calculated by applying the flagtool command in SAMtools.

Transcript abundance and differentially expressed genes

Raw read counts for each transcript were calculated using Htseq-count and normalized to transcripts per million (TPM). TPMi = (Ni/Li)*106/sum(Ni/Li+……+Nm/Lm); Ni represents the reads mapping to the i-th gene, and Li represents the total length of the exons of the i-th gene. DEGs in the different treatments were analyzed using R (DESeq2). Raw counts were fed to DESeq2, and only those genes in which the —log2 (fold change)— was greater than 1 and the false discovery rate was less than 0.01 were identified as DEGs. The expression patterns of the DEGs were made visible using R pheatmap. Differentially expressed transcription factors (TFs) were predicted by submitting the DEGs to the PlantTFDB 4.0 database (http://planttfdb.gao-lab.org/) (Jin et al., 2017).

GO analysis was carried out for the DEGs using the agriGO database (http://systemsbiology.cau.edu.cn/agriGOv2/index.php), and the P values were corrected to control falsely rejected hypotheses during the GO analysis. GO annotations of 23,397 genes from the genome of C. equisetifolia were taken as the reference set, and GO annotations of DEGs were taken as the test set. DEGs were classified and analyzed statistically according to three major functional modules, namely molecular function, biological process, and cellular component, and functional annotation of the DEGs was conducted according to these three module parameters. Paralogs and orthologs were identified by running a BLASTN (Altschul et al., 1997) for all nucleotide sequences for each species, based on the same method described by Blanc & Wolfe (2004).

Quantitative reverse-transcription polymerase chain reaction

Total RNA samples used in transcriptome sequencing were also used for qRT-PCR. Reactions were performed on an Applied Biosystems 7500 Real-Time PCR using a SYBR Premix Ex Taq™ kit (TaKaRa, Japan) following the manufacturer’s instructions. The combination CaeUBC and CaeEF1 α was used as an internal control (Fan et al., 2017). Primers were designed using Primer Premier ver. 5.0 to allow for amplification of 80–200 bp products. Gene names, sequences, and the primers used for qRT-PCR analysis are listed in Table S1. Thermal cycling conditions were 30 s at 95 °C followed by 40 cycles of 5 s at 95 °C and 34 s at 60 °C. A dissociation curve was obtained by heating the amplicon from 60 °C to 95 °C. Each sample was analyzed at least three times. Standard curves were established for all genes investigated using a series of amplicon dilutions. Relative expression level was calculated using the 2−ΔΔCT method (Schmittgen & Livak, 2008).

Availability of data and materials

Raw Illumina sequence data were deposited in the Short Read Archive of the NCBI database (project accession number SRP064226).

Results

Morphological changes in roots

We previously studied C. equisetifolia clone A8 under different NaCl concentrations (Fan et al., 2017). The number of lateral roots was decreased under 200, 400, or 600 mM NaCl treatment, while etiolated and wilted leaves and black and decayed roots were observed under 400 mM and 600 mM NaCl treatments. To gain more accurate insight into the salt response mechanism and identify DEGs associated with salt stress rather than senescence or death, we performed the salt stress treatment of C. equisetifolia under 200 mM NaCl.

As the duration of exposure to stress increased, so did the extent to which root growth and formation of lateral roots were inhibited. Grey roots (white boxes) were also observed at 24 and 168 h (Fig. 1), and no nodule formation was observed. Similarly, Ngom et al. (2016) reported that nodule formation did not occur in seedings inoculated with Frankia Ccl3 or CeD strains, at NaCl concentrations above 100 and 200 mM, respectively. Root microstructure in the cortex, vascular system, and aerenchyma cells was almost similar under 0 h and 1 h treatments (Fig. 2). However, cell disfiguration in the epidermis became apparent after 6 h. Some epidermal cells were shed under the 24 h treatment, with slight plasmolysis in some cortex cells; in addition, the pericycle cells continued to shrink, and the epidermal cells became separated from the cortex cells. After 168 h of treatment, the roots shriveled even further and became disfigured. Epidermal cells became detached, and more aerenchyma tissue was produced by the cortex cells under salt stress (Fig. 2). The epidermal cells were thickened considerably and were tightly packed together; the endo-epidermis, which forms a thicker barrier, protects the aerenchyma from damage. The protective gap produced between cortex cells and pericycle cells was widest at 168 h of treatment, and the cells neighbouring the pericycle cells were thicker, forming a second barrier created by the root.

Figure 1 Morphological changes in roots of C. equisetifolia under salt stress.

The seedlings were treated by 200 mM NaCl solution for 0, 1, 6, 24, and 168 h. Root analysis was performed in C. equisetifolia in response to salt stress. The white arrows indicate new roots, and the white boxes indicate dead roots.

Figure 2 Microstructure changes in roots of C. equisetifolia under salt stress.

After treatment with 200mM NaCl solution for 0, 1, 6, 24 and 168 h, roots were collected for microstructure analysis. Ep, Epidermis; Ex, Exodermis; En, Endodermis; St, Stele.

Ultrastructural changes in roots

To analyze the changes in cell structure in roots of C. equisetifolia in response to salt stress in greater detail, we also performed ultrastructural analysis. Because the changes observed under 1 h and 6 h treatments were similar, our analysis focused on the 6 h treatment. As shown in Fig. 3, epidermal cells with abundant accumulation of cytoplasm around the cell wall were closely connected in control plants. With increasing duration of exposure to salinity, marked changes in the shapes of epidermal cells and cortical cells appeared (Figs. 3 and 4). The number of mitochondria increased in cortical cells and stele cells under the 24 h treatment but not under the other treatments (Figs. 4 and 5). Nevertheless, no significant plasmolysis was observed in stele cells (Fig. 5). After 168 h of NaCl treatment, plasmolysis was evident and the cell wall of the epidermal cells became loose; some exterior parts of the cell wall were isolated and desquamated, the mitochondrial membrane structure showed deterioration, and there was slight disorganization of the matrix and cristae in epidermal cells and cortical cells (Figs. 3 and 4). However, stele cells with intact membranes and organelles including the cell wall, endoplasmic reticulum, mitochondria, nucleus, and vacuole were also observed (Fig. 5). In addition, cessation of cell growth and inhibition of cell division were also evident, as indicated by the number of cells and accumulation of biomass in the pericycle cells. These results indicate that C. equisetifolia initiates an adaptive response to salinity stress in its roots.

Figure 3 Ultrastructure of epidermis change in root cells of C. equisetifolia under salt stress.

Root analysis was performed in response to salt stress by 200 mM NaCl solution for 0, 1, 6, 24, and 168 h. The first line of the picture: the changes of the local epidermis at different time periods under salt treatment. The second line of the picture: individual epidermal cells treated by salt at different times. The third row of the picture: the number of mitochondria in a single epidermal cell during different time periods under salt treatment. The fourth row of the picture: the changes of mitochondrial structure in a single epidermal cell during different time periods under salt treatment. CW, cell wall; M, mitochondria.

Figure 4 Ultrastructure of cortex change in root cells of C. equisetifolia under salt stress.

Root analysis was performed in response to salt stress by 200 mM NaCl solution for 0, 1, 6, 24, and 168 h. The first line of the picture: the changes of the local cortex at different time periods under salt treatment. The second line of the picture: individual cortical cells treated by salt at different times. The third row of the picture: the number of mitochondria in a single cortical cell during different time periods under salt treatment. The fourth row of the picture: the changes of mitochondrial structure in a single cortical cell during different time periods under salt treatment. CW, cell wall; M, mitochondria.

Figure 5 Ultrastructure of stele change in root cells of C. equisetifolia under salt stress.

Root analysis was performed in response to salt stress by 200 mM NaCl solution for 0, 1, 6, 24, and 168 h. The first line of the picture: the changes of the local stele at different time periods under salt treatment. The second line of the picture: individual stele cells treated by salt at different times. The third row of the picture: the number of mitochondria in a single stele cell during different time periods under salt treatment. The fourth row of the picture: the changes of mitochondrial structure in a single stele cell during different time periods under salt treatment. CW, cell wall; M, mitochondria; Pe, pericycle; Ve, vesicle; V, vacuole.

Changes in ions content

With the increasing of salt stress, the content of sodium and chloride ions in C. equisetifolia roots increased, with the content of chloride ions obviously higher than that of sodium ions. When the treatment time reached 168 h, the content of Cl− was as high as 59.903 g kg−1 and the content of Na+ was 33.50 g kg−1. Compared with the control group, the content of potassium decreased slowly between 1 h (14.71 g kg−1) and 24 h (10.14 g kg−1) after initiation of salt treatment, while the content of potassium was significantly lower after 168 h (1.63 g kg−1) of salt treatment (Table 1). In addition, the K+:Na+ ratio showed a clear decrease with salt treatment time. In the control group, the K+:Na+ value was 3.12; after 1 h of treatment, the ratio was 1.15; with increasing treatment time, the ratio continued to decrease, reaching 0.05 at 168 h.

Alignment and assembly of RNA-Seq datasets

We performed transcriptome analysis based on high-throughput RNA-Seq. In total, 561,652,970 clean reads were generated, and the number of clean reads per library ranged from 68,520,708 to 94,889,166. The clean reads were mapped to the C. equisetifolia reference genome (Ye et al., 2019) with mapping rates ranging from 77.06% to 81.52% (Table S2). Scatter plots of data from all samples showed that the samples formed two clusters: samples from the 1 h and 6 h treatments were clustered in one group and showed a relatively close relationship with the control (0 h), thus representing the early stages of the response, whereas samples from the 24 and 168 h treatments were clustered in a second group, representing the later stages of response (Fig. S1A). Pair-wise values of Pearson’s correlation of expression between biological replicates ranged from 0.985 to 0.999 (Fig. S1B), and correlations of expression values between treatment and control samples indicated that the early-stage samples were closely related, with only a moderate difference (R = 0.792 for 1 h vs 0 h and R = 0.723 for 6 h vs 0 h) (Fig. S1C). The scatter of gene expression values and the low correlations (R = 0.650 and R = 0.339) revealed that the number of DEGs changed under the 24 h and 168 h treatments. These results indicate that the late-response stage was not simply the result of repression but also involved activation of new groups of genes associated with salt stress.

Identification of DEGs and gene enrichment analysis under salt stress

DEGs were also detected as exposure to stress became more prolonged. A total of 10,738 DEGs were identified (Table S3-1): 2399 in the 1 h treatment, 5668, at 6 h, 7660 at 24 h, and 6849 at 168 h compared with the control (0 h) (Fig. S2 and Table S3-2 to Table S3-5). There were 523, 499, 1535, and 1349 DEGs specific to each time point, whereas 1103 DEGs were common to all the four salt treatments (Fig. 6 and Table S3-6 to Table S3-11). This result indicates that complex transcriptional regulatory events occurred during the later stages of the salt treatment.

Table 1 Ions content changed in C. equisetifolia root under salt stress.

Na+, Cl− and K+ content in root under 200 mM NaCl treatment for 0, 1, 6, 24, and 168 h. Values represent mean ± standard deviation; n = 3. Values within a column with different letters indicate significant difference (P < 0.05 using analysis of variance at 95% confidence level).

Name	K+/Na+	K+ (g/kg)	Na+ (g/kg)	Cl− (g/kg)	
Control	3.123a	15.118 ± 1.057a	4.841 ± 1.257a	8.565 ± 1.254a	
H1	1.149b	14.71 ± 0.32a	12.798 ± 0.132b	15.654 ± 1.798b	
H6	0.782bc	13.618 ± 1.118a	17.416 ± 1.829c	23.416 ± 0.909c	
H24	0.328bc	10.14 ± 0.657b	30.903 ± 1.404d	48.237 ± 3.208d	
H168	0.049c	1.634 ± 0.161c	33.503 ± 1.352d	59.903 ± 2.024e	

Figure 6 DEGs analysis at different time points under salt stress.

Venn diagram showing the number of DEGs in C. equisetifolia at 1, 6, 24, and 168 h of exposure to 200 mM NaCl solution. The column diagram indicated the number of up-regulated and down-regulated DEGs. The table showed the DEGs between the two samples.

GO enrichment using a P value of ≤ 0.05 as the cut-off identified 401 GO terms enriched during the entire duration of salt stress (Table S4-1). A large number of DEGs were associated with biological processes (209 terms) and molecular functions (157 terms) under salt stress. The enriched categories in C. equisetifolia roots consisted of genes involved in signaling, transport, metabolism, regulation, and development. To gain further insight into the biological processes associated with the observed temporal changes, GO enrichment analysis was performed at each of the time points (Table S4-2 to Table S4-5). Many categories associated with signal transduction and DNA were enriched in the 1 h and 6 h treatments. The biological processes represented by these ontologies were significantly less enriched during the later stages of salt stress (Fig. 7 and Table S4-2 to Table S4-5). This implies that salt stress triggered numerous signal transduction pathways and DNA replication and repairs processes within 1 h and 6 h of the onset of salt stress. Salt stress induced the responses from multiple hormones in C. equisetifolia, including ABA, JA, gibberellin, and auxin (Fig. 7). Stress-response ontology terms enriched at 1 h and/or 6 h indicated that the roots of C. equisetifolia can respond immediately to salt stress by activating stress-response genes. It is also noteworthy that some ontologies representing transport, cell development, and growth were enriched at 24 and/or 168 h (Fig. 7), and that DEGs identified at the later stages (24 and 168 h) were specifically enriched in ontologies associated with death, corresponding to root cell apoptosis in the earlier experiment (Fig. 7). Both oxidation and detoxification GO terms, which are associated with eliminating ROS induced by the stress, were enriched throughout.

Figure 7 GO enrichment analysis at different time points under salt stress.

The biological processes analysis of differentially expressed genes (DEGs). Log10 was applied to the number of enriched DEGs. The darker the color, the more DEGs are enriched.

Response of DEGs to salt stress in C. equisetifolia

To obtain an overall view of the expression profiles of the 10,738 DEGs identified in C. equisetifolia, we constructed a heat map using R (Pheatmap) (Table S5). The transcriptome responses of C. equisetifolia treated with salt stress for different lengths of times are shown in Fig. 8A. Oxidase superfamily proteins, protein kinases, MAPK signaling cascades, ion homeostasis, and other related genes were identified during this process. All DEGs were divided into four clusters based on their expression patterns. Cluster 1 comprised MAPK4 (CCG019781), calcium-transporting ATPase (CCG010076 and CCG005018), HKT1 (CCG006526 and CCG006527), and CHX (cation/hydrogen exchanger) (CCG009112 and CCG013892), which were quickly induced within 1 h of treatment and responded to salt stress immediately. Cluster 2 comprised CIPK21 (CBL protein-interacting protein kinases) (CCG001663), CDPK2 (calcium-dependent protein kinase) (CCG001182), VQ motif-containing (CCG005740 and CCG004208) and GST25 (glutathione S-transferase) (CCG015479, CCG015480, and CCG015481), which were induced at 6 h and remained upregulated as the duration of stress increased. Both these above clusters also included NHX (sodium hydrogen exchanger) genes such as NHX1 (CCG027771 and CCG003145) and NHX2 (CCG028406). Cluster 3 comprised SOS2 (CCG023938), MAPK9 (CCG029210), GST (CCG013823) and CDPK1 (CCG013990), which were upregulated at 6 h of treatment and peaked at 24 h. It is noteworthy that Cluster 4 comprised HAK5 (CCG012758), KUP6 (CCG017938), and some genes related to cell death (CCG014816, CCG014814 and CCG013802), the expression of which was specifically induced under the 168 h treatment (Figs. 9A and 9B). Several genes were selected for validating the transcriptome data using qRT-PCR analysis, and these showed similar expression patterns to those indicated by TPM values (Fig. S2).

Figure 8 DEGs at different time points under salt stress.

(A) Expression profiles of all DEGs at different time points under salt stress. Log10 was performed on the TPM value. The color scale on the right side represents values of normalized TPM values. Blue represents low expression and red indicates a high expression level. The heatmap was constructed by R package (Pheatmap). (B) The distribution of representative 689 TFs. Different colors represent different TFs.

Figure 9 Expression profiles of DEGs related to ion transport and PCD-related genes at different time points under salt stress.

(A) Expression pattern of 26 DEGs related to ion transport. The color scale on the right side represents values of normalized TPM values. Blue represents low expression and red indicates a high expression level. The heatmap was constructed by R package (Pheatmap). (B) Expression profiles of PCD-related genes. The color scale on the right side represents values of normalized TPM values. Blue represents low expression and red indicates a high expression level. The heatmap was constructed by R package (Pheatmap).

We also identified TFs differentially expressed in response to salt stress. A total of 689 TFs were distributed among 44 families in C. equisetifolia (Table S6). The majority of TF genes belonged to the bHLH, MYB, NAC, AP2/ERF, WRKY, bZIP, HD-ZIP, GRAS, and C2H2 families (Fig. 8B). We identified 99 MYB, 44 NAC, 75 AP2/ERF, and 43 WRKY genes, which function under salt stress and also under biotic stress (Table S6). Among TF families, the number of genes belonging to the MYB superfamily was the largest, similar to the potential transcriptional regulatory factors in the symbiosis of Casuarina and Rehmannae radiosurface (Diédhiou et al., 2014). For example, MYB2 (CCG011536), which is involved in the induction of salt-responsive genes that are induced by ABA (Dubos et al., 2010), and WRKY70 (CCG020989), which modulates tolerance to osmotic stress by regulating stomatal aperture (Jing et al., 2013), were also induced under salt stress in roots of C. equisetifolia. Additionally, most AP2/ERF, WRKY, and bZIP genes grouped into Cluster 2, which were up-regulated during the later stages of salt treatment. It should also be noted that LBD, GRAS, ARF, and GRF genes involved in plant growth and development showed differential expression in this study. For example, Scarecrow-like, a GRAS protein (CCG016614), and zinc-finger protein 5 (CCG017923), which mainly control the coordination of root cell elongation and development (Heo et al., 2011; Xie et al., 2019), also responded to salt stress in C. equisetifolia. We speculate that these genes contribute to the response to salt stress either directly or by resisting negative effects through regulating growth and development under long-term salt stress.

Discussion

Salinity is one of the extreme environments that limit plant growth. C. equisetifolia, which can tolerate salinity, is used for creating shelter forests in coastal belts. However, few studies have examined the mechanism of its adaptation to salt sress in detail. In the present study, although 200 mM NaCl did not affect the growth of C. equisetifolia seedlings substantially, longer exposure to salinity inhibited root growth and the formation of new lateral roots; the roots also turned dark, cells were shed, and epidermal cells showed plasmolysis (Fig. 1)—symptoms that have been reported in other plant species as well (Tu et al., 2014). With increasing duration of salt treatment, the structure of root cells changed, but remained intact. Most of the cells in the outer epidermis died and were shed, which was a different response from that seen in other plants such as rice (Céccoli et al., 2011).

We identified fewer DEGs at 1 and 6 h compared with 0 h than at 24 and 168 h, as ascertained through transcriptome analysis (Fig. 6). Histological examination of the corresponding tissues showed that morphological changes became more obvious with increasing duration of salt treatment (Fig. 2). GO enrichment analysis annotated genes associated with signaling, stress response, hormone, and transport ontologies during the early stages of salt stress (1 h and 6 h) (Fig. 7). This suggests that genes related to signaling, transport, hormones and to stress responsees are initiated immediately in roots of C. equisetifolia exposed to salt stress. Earlier studies reported similar responses, namely rapid and dynamic changes in root and shoot growth in plants exposed to salinity (Passioura & Munns, 2000; Munns, 2002). However, ultrastructural analysis showed slight plasmolysis in epidermal cells after 6 h of exposure to salt. These initial changes in growth were driven by the osmotic component of salt stress, which immediately affects the water status of the plant, preventing cell elongation. Within several hours of treatment, partial recovery of growth occurred owing to the uptake of inorganic ions and the biosynthesis of compatible osmolytes, which reduce the water potential of cells until cell expansion can resume (Yu et al., 2013).

It is particularly noteworthy that under the 24 and 168 h salt treatments, DEGs were enriched for metal ion and sodium ion transport GO terms (Fig. 7). We identified 26 DEGs related to ion transport (Fig. 9A). Plants are known to adapt to salt stress through the SOS pathway to maintain ion balance in cells (Zhu, 2003; El Mahi et al., 2019). Of the 26 DEGs, SOS2 was up-regulated at 6 h and reached its maximum expression at 24 h, and NHX7 (SOS1) genes responded positively to salt stress. Some TFs such as bZIP, which activates the expression of genes involved in cytoplasmic ion homeostasis, such as the Na+ transporter HKT1 and the Na+/H+ anti-transporter SOS1 in Arabidopsis (Yang et al., 2009), were also expressed in C. equisetifolia roots under salt stress. Two HKTs were identified among the DEGs (Fig. 9A). Even more noteworthy is the fact that sodium and chloride content increased significantly after 24 h of salt treatment (Table 1). Maintaining potassium balance is an essential part of plant response to salt stress. AtHKT1 regulates K+ state (Wang et al., 2018), whereas HvHKT1;5 in stele cells, negatively regulates salt tolerance in barley (Huang et al., 2020). Overexpression of OsHAK5 increases the K+/Na+ ratio and tolerance to salt stress in rice seedlings (Yang, Zhang & Hu, 2014). In the present study, the expression of most members of the HAK genes family was up-regulated and potassium concentration decreased slowly between 1 h and 24 h after initiation of salt treatment. These results imply that HAK is involved in maintaining K+/Na+ homeostasis in response to salt stress in C. equisetifolia. Taken together, our results suggest that ion transport plays an important role in the response of C. equisetifolia to long-term salt stress. Two HKTs genes were identified among the DEG, and some HAK genes were highly expressed after 168 h of salt treatment. HKT and HAK genes can therefore be used as candidates to study the molecular function of salt tolerance in C. equisetifolia.

It is well known that plants accumulate more ROS under salt stress. The DEGs associated with salinity treatment were greatly enriched in GO terms related to ROS-related biological processes and molecular functions (Table S4), such as the hydrogen peroxide catabolic process and the oxidation–reduction process. Heterologous expression of GhWRKY41 in Nicotiana benthamiana was reported to enhances salt tolerance by regulating ROS scavenging (Chu et al., 2015). Furthermore, bHLH92 (CCG027461) and WRKY33 (CCG011999 and CCG003169) play a regulatory role in ROS detoxification through glutathione S-transferases and peroxidase (Miller et al., 2010), suggesting that these TFs mediate ROS scavenging and oxidative stress-induced signaling pathways. Additionally, ROS might act as a signal molecule controlling plant programmed cell death (PCD) (Gechev & Hille, 2005; Petrov et al., 2015), and salt treatment is known to induce PCD in root tips (Chen et al., 2009) and leaves (Ambastha et al., 2017) in rice. Salt stress induces an increase in ROS before PCD in tobacco protoplasts, pointing to an association between oxidative damage and PCD (Lin, Wang & Wang, 2006). In the present study, GO terms associated with ROS-related biological processes were enriched throughout the salinity treatment, whereas PCD-related genes were enriched among DEGs during the later stages (Fig. 7). Therefore, we conclude that generation of ROS is activated under salt stress, initiating PCD, which is important for regulating the response of C. equisetifolia to salt stress.

Ultrastructural analysis revealed significant changes in cell structure after168 h of salt stress (Fig. 2). Correspondingly, DEGs identified at the late time points (24 h and 168 h) were specifically enriched in ontologies associated with cell death, indicating that root cell apoptosis is part of the response to salt stress (Fig. 7). PCD is an important part of the response to salt stress, ensuring the plant has enough time to activate mechanisms for adapting to stress. Under non-lethal conditions, PCD induced in severely salt-stressed roots removes most of the salt-susceptible cells, which are subsequently replaced with cells better adapted to the stress (Kacprzyk, Daly & Mccabe, 2011). In rice, root cell death under salinity starts from epidermal and cortical cells and progresses to the endodermis and stele to minimize the adverse effects of stress. The dead cells prevent the influx of excess salt ions into the stele and into shoots, leading to larger amounts of salt being excluded (Liu et al., 2006). Similar results were also obtained in the present study: epidermal cells were shed and a more compact layer of cells formed after 168 h of salt treatment (Fig. 2). After 24 h and 168 h of treatment, plasmolysis was seen in both epidermal cells and cortical cells, and mitochondria were seriously damaged (Figs. 3 and 4). Temporal trends in enrichment of GO terms among DEGs corresponded closely with observed changes in root morphology in response to salt stress. Furthermore, 15 PCD-related genes were identified as thaumatin-like proteins (TLPs), most of which were upregulated at 168 h (Fig. 9B). TLP genes can be induced by salicylic acid (SA) or JA hormone signaling, thus playing an important role in plant stress defense processes (Rout et al., 2016; Sun et al., 2020). Lopes et al. (2019) reported that TLP genes have selective anticandidal activity, inducing apoptosis via a membrane receptor (Lopes et al., 2019). DEGs identified at 168 h were specifically enriched in ontologies associated with the SA metabolic process and JA- mediated signaling pathways. These results showed that TLP expression is regulated by SA and JA in C. equisetifolia under salt stress, leading to PCD. TLP genes can also be used as candidates for studying the molecular function of salt tolerance in C. equisetifolia.

Previous studies have shown that C. equisetifolia tolerance to high salt concentrations is innate (Scotti-Campos et al., 2016; Selvakesavan et al., 2016) and that in vitro salt tolerance of Frankia strains has no correlation with the salt tolerance of C. equisetifolia under salt-stressed conditions (Ngom et al., 2016). Furthermore, nitrogenase activity in nodules is insignificant at 200 mM NaCl (Duro et al., 2016a; Duro et al., 2016b; Mansour et al., 2016). Similarly, 100 mM NaCl concentration has a significant inhibitory effect on nodule function in Elaeagnus commutate (Shao, Markham & Renault, 2020). Previous studies have revealed that salt tolerance in C. glauca is linked to photosynthetic, primary metabolic adjustments and to an effective antioxidant machinery (Graça et al., 2019; Batista-Santos et al., 2015; Jorge et al., 2019). Our research results showed that the sodium content of C. equisetifolia roots was significantly increased (30.903 g kg−1) after 24 h of salt treatment. C. equisetifolia can sequester Na+ in root tissues to prevent sodium transfer to the shoot (Fan et al., 2017), implying that HKT, HAK, and NHX are involved in maintaining K+:Na+ homeostasis in C. equisetifolia in response to salt stress. Structural analysis revealed more obvious deformation of the cell membrane with increasing duration of salt stress (Fig. 2). Meanwhile, 15 PCD-related genes were induced by SA or JA to participate in the salt stress response. Interestingly, selection of appropriate fungal strains is crucial for improving C. equisetifolia performance in saline soils (Djighaly et al., 2018). The effect of Frankia symbiosis on salt tolerance of C. equisetifolia will be our next research focus.

Conclusion

Soil salinity is a severe environmental constraint on plant growth. Roots of C. equisetifolia were exposed to 200 mM NaCl solution for 0, 1, 6, 24, and 168 h. Epidermal cells sloughed off and a more compact layer of cells formed after 168 h of treatment, while potassium concentration remained relatively stable. Ultrastructural analysis revealed cell deformation and mitochondrial damage in the epidermis and endodermis but less damage in stele cells. A total of 10,378 DEGs were identified through transcriptome analysis. Oxidative stress and detoxification increased throughout the treatment period, and expression of genes related to these processes was upregulated. Salt stress led to higher Na+ content in C. equisetifolia roots. We determined that, in order to prevent excessive accumulation of Na+, which is toxic to cells, some genes identified in the present study, including those encoding Na+/H+ transporters, K+ transporters, and potassium channel proteins, were upregulated in response to salt stress. As stress continued, specific ontologies associated with cell death and PCD were enriched among DEGs, and genes related to these processes were significantly upregulated at 168 h of salt treatment. Some TFs, such as those belonging to the WRKY and the MYB gene families, were induced under salt stress. In the future, we will focus on candidates including ion transporter-related genes and PCD-related genes and verify their molecular functions using plant transformation.

Supplemental Information

Supplemental Information 1 The scatter-plot analysis of samples with two biological replicates (rep_1 and rep_2)

(A) All samples were cluster. (B) Pair-wise Pearson’s correlations of expression values between biological replicates. (C) The correlation of expression values between the treatment and control samples.

Click here for additional data file.

Supplemental Information 2 The expression pattern of some DEGs and validation of RNA-seq data

Left panel: The relative expression levels of target genes from qRT-PCR results, and TPM values were acquired by RNA-seq. Right panel: Expression profiles of some DEGs, and the heatmap was constructed by R package (Pheatmap).

Click here for additional data file.

Supplemental Information 3 Primers used for qRT-PCR of genes in C. equisetifolia

Click here for additional data file.

Supplemental Information 4 Mapping rate of clean reads to the reference genome

Click here for additional data file.

Supplemental Information 5 Summary of the DEGs

The total differentially expressed genes in Table S3 −1. The DEGs of 1, 6, 24, and 168 h vs control are detailed in Table S3 −2 to Table S3 −5. The DEGs of different salt treatment are detailed in Table S3 −6 to Table S3 −11.

Click here for additional data file.

Supplemental Information 6 Gene ontology of the DEGs

Gene ontology of total DEGs in Table S4 −1. Gene ontology analysis at each of the time points are detailed in Table S4 −2 to Table S4 −5.

Click here for additional data file.

Supplemental Information 7 The TPM values of the total DEGs

Click here for additional data file.

Supplemental Information 8 The DEGs encoding TFs under salt stress in C. equisetifolia

Click here for additional data file.

Supplemental Information 9 Raw data

Ions content changed in C. equisetifolia root under salt stress and the relative expression levels of target genes from qRT-PCR results.

Click here for additional data file.

We acknowledge Professor Mengzhu Lu’s suggestions about the experimental design. We also thank Dr. Min Li for his critical reading of this manuscript.

Additional Information and Declarations

Competing Interests

Author Contributions

DNA Deposition

Data Availability

The authors declare there are no competing interests.

Yujiao Wang performed the experiments, analyzed the data, prepared figures and/or tables, authored or reviewed drafts of the paper, and approved the final draft.

Jin Zhang, Bingshan Zeng and Chonglu Zhong conceived and designed the experiments, authored or reviewed drafts of the paper, and approved the final draft.

Zhenfei Qiu performed the experiments, analyzed the data, prepared figures and/or tables, and approved the final draft.

Yong Zhang analyzed the data, authored or reviewed drafts of the paper, and approved the final draft.

Xiaoping Wang and Jun Chen performed the experiments, prepared figures and/or tables, and approved the final draft.

Rufang Deng analyzed the data, prepared figures and/or tables, and approved the final draft.

Chunjie Fan conceived and designed the experiments, analyzed the data, prepared figures and/or tables, authored or reviewed drafts of the paper, and approved the final draft.

The following information was supplied regarding the deposition of DNA sequences:

Ten samples were collected and labelled as follows to reflect the duration of stress and replication: Ck-1 and Ck-2 (Ck for control or check) and 1h-1, 1h-2, 6h-1, 6h-2, 24h-1, 24h-2, 168h-1 and 168h-2.

The sequences are available at the Short Read Archive of the NCBI database: SRP064226.

The following information was supplied regarding data availability:

Raw data for correlation analysis of DEGs are available in the Supplemental Files.

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
