# Peer review of "Transcriptome and structure analysis in root of Casuarina equisetifolia under NaCl treatment"

_PeerJ, doi:10.7717/peerj.12133_

## Round 0.1 · original submission · Major Revisions

All three reviewers agree that the topic of your manuscript is relevant and your net contribution to it would be worthy.

However, the reviewers and myself also agree that a very extensive revision of the manuscript is needed, as far as English, data presentation (including the quality of the figures), conclusions drawn and their discussion, references cited, and many other points, as detailed in the reviewers' reports.

In addition, there is concern about the statistical analysis of your results.

A cover letter where you address point by point all of the reviewers' suggestions and comments will have to accompany your revised manuscript, where all the changes should be clearly highlighted.

Reviewer 1 ·

Basic reporting

The article of Wang et al. addresses an important topic under the context of saline agriculture that will cover more than 50% of the agricultural land within few decades. Thus, understanding the mechanisms underlying plant resilience to salt is of utmost importance to develop appropriate stress mitigation strategies. Casuarina equisetifolia belongs to the group of actinorhizal trees which are distributed worldwide and are able to thrive in a wide range of extreme environments. Together with legumes, these plants belong also to the restricted group of plants that are able to establish root-nodules with N-fixing bacteria, reinforcing their importance in plant research.

Experimental design

The work is original and the research strategy and the experimental design have been carefully drawn. The methods are well described.

Validity of the findings

All data are provided, robust and statistically sound. However, there are some aspects that should be improved (see comments below).

Additional comments

Although the research strategy and the experimental design has been carefully drawn and is clear and concise, overall the MS has major flaws that should be overcome before publication.

(1) The Language should be fully edited, preferably by specialized services or by a fluent speaker

(2) The Introduction should include a chapter with the state of the art of the research developed in actinorhizal plants, particulary in the casuarinaceae. Actually, this topic is not included, neither in the introduction nor in the discussion and from the ca. 45 cited publications only 6 concern casuarina research (4 of which are self-citations). I have included in point (6) a list of bibliography that, in my opinion, should be included to enrich the extent of the paper.
(3) In the Materials and Methods sections, please explain:
- the characteristics (in relation to salt stress) of the genotype used (selection criteria)
- the choice of 200 mM NaCl (particularly taking into account that in the introduction 500 mM are referred, it is not clear why did the study was conducted with 200 mM)
- why was hyrdroponic culture used instead of sandy soils, where C. equisetifolia occurs naturally?
- why was the impact of the symbiosis with Frankia excluded?
(4) The results are clear and concise.
(5) The Discussion is focused on anti-transporters and PCD, rather than in the four gene clusters and their respective interactions to promote stress tolerance; also, it is not clear if the genotype used in this study is tolerant, moderately tolerant or susceptible; a deep comparison with other Casuarina spp., actinorrhiza as well as halophyte species, should be mandatory to increase the relevance of the research conducted under this study, clearly highlighting the novelty of this work; equally important is to discuss the putative advantages (or not) of the symbiosis with Frankia and mycorrhiza.
(6) Relevant bibliography to be included:
10.3389/fsufs.2020.601004
10.1007/s11738-020-03088-y
10.3390/ijms21010078
10.1016/j.envexpbot.2019.103808
10.1007/s11306-017-1234-7
10.1007/s13199-016-0386-y
10.1007/s13199-016-0425-8
10.1007/s13199-016-0424-9
10.3389/fpls.2016.01331
10.1016/j.jplph.2016.03.012
10.1007/s11104-015-2666-3
10.1016/j.plaphy.2015.07.021
10.1111/j.1399-3054.2006.00633.x
10.1111/j.1747-0765.2006.00005.x
https://www.jstor.org/stable/23616578?seq=1

Reviewer 2 ·

Basic reporting

See below

Experimental design

See Below

Validity of the findings

See beow

Additional comments

The manuscript of Wang et al describes an investigation into the salinity resistance of Casuarina equisetifolia via a transcriptome approach. Building off the released C. equisetifolia genome sequence, the system is ripe for global expression studies. However, there are number of flaws or absence of information that affects this manuscript. Some of these problems are significant or others may be explained or rationalized but are not in the text. In general, reads well and the data presented clearly.

1. The RNASeq data as all high-throughput sequence data to needs to be submitted a repository for public access via NCBI GEO. This requires metadata and actual data submission. You must write about this fact in a data deposition statement. It must be access via the public for validation of these results. This information is not present. However, the raw reads are available at NCBI, but the GEO information should also be deposited and described in text. The statement on raw reads should also be moved to the Material and Methods section under a separate section.
2. Causuarina equisetifolia and C. glauca that are in symbiosis with Frankia are more tolerant to salt effects than uninnocualted plants (Ngom et al 2018). Why were these studies done with uninoculated plants? Explain?
3. How do you know what is a salt effect and not an osmotic effect? Were any controls run to eliminate that possibility?
4. There is a lot of literature that is missing in the introduction or discussion on Casuarina and salinity effect For example Ng0m et al 2018; Craco et al 2020 and others. A quick review of all of the literature would help this study

Reviewer 3 ·

Basic reporting

The manuscript in english, which nedds Some improvement. Some sentences are unclear.
TITLE : Title should be changed. There is no elucidation of mechanism (no functional approach); It is just e description of genes regulated. last there is no real link done between the histology and the transcripto.
ABSTRACT: Conclusion should be modified

Introduction : Litterature is not recent enough. Intrduction should be reconsider by including more recent bibliography. There are lots of recent review and data on salt stress responses in plant.
Last the introduction focus on a list of genes ...but does not cite other transcriptomics studies neither the gene families that were modulated during salt stress. It should be added to compare if Casuarina equisetifolia has a similar response or not regarding salt stress.
See : Park etal 2016 Mol Cell Biol
Yu et al. 2020 TIPS
Farhat et al., 2019
van Zelm et al ARPB 2020
Morton et al 2019 Plant J
Liu et al 2019 BMC PB on populus (an other tree)

Structure of the manuscript is standard.

Figures are not good. Often too small and legends are not precised enough. See detailes comments.

Experimental design

Research question is unclear. Salt stress mecanisms in casurina equisetifolia are not precisely elucidated and not compared to not tolerant species. Thus difficult to see what makes this species more tolerant.
Considering technics, there are some weakness; The study is done with only 2 biological replicates (transcriptomics) and this is not the actual standard which requires 3 biological replicates.
Statistics are somhow inapropriate and should be reconsidered regarding to the low number of samples (non parametric tests).
I am also surprised of the sample clustering; Ah close to 6h = OK; 24h clustering with 168h (7days)...I am quite surprised. Thsi could be due to the low number of replicates and to a weakness of statistic methods; Please you may check.

Validity of the findings

The subject is oruiginal but should be more exploited in the discussion. What makes casuarina equisetifolia more tolerant? Comparison with other study. Other works done on Casuarinacea and salt stress should be considered (Ngom et al, 2016 FIPS; Djighaly et al, 2018 annals of forest science). There are data about casuarina in salted condition and authors should see if their observation are similar or not. There is a strong part focusing of transcription factors and the paper of Diédhiou et al. BMC Plant Biology 2014 is not cited, although it is a study of FT in actinorhizal species (Casurina and alnus).
Papers of gaca et al Int J Mol Sci. 2019 should also be considered, soame for Scotti-Campos P, J Plant Physiol. 2016
All these paper deals with salinity and casuarina;and my give some interesting comparison sources.
In brief, results should be more analysed, and result part of the paper reorganised.
Discussion and conclusion should include more recent litterature and try to do a comparison of transcriptomic data and histology results; Did you find gene inviolved in cell remodelling ?

Additional comments

The research is of novelty; However the main default is that there is only 32 biological replicates for transcriptomics which does not respect the actual standard (3). Then staistical anlysis is weak. Lots of litterature on saline stress in plants and in casurinaceae is missing.
Detailes comments are provided as a separated file.
Study nedds to be reconsidered to fit the standard for publication;
Review of the manuscript 2021:02:58311:0:1:NEW 4 Mar 2021
Comprehensive elucidation of the mechanism of salinity resistance by cell structure and transcriptome analysis in root in Casuarina equisetifolia
Tittle
Line 1 : Title should be modified to reflect the study.
Introduction
Line 52 to 77 This part lack of recent references on salt stress in plant and on transcriptomic studies done on this problematic. What are mechanisms of tolerance? Which sets of genes are implicated?
Then it will be easier to compare (in the discussion part) and to explore what makes Casuarina equisetifolia more tolerant. This is the main question of the paper and it does not show up.

Papers to be considered (among other):
Park et al 2016 Mol Cell Biol
Yu et al. 2020 TIPS
Farhat et al., 2019
van Zelm et al ARPB 2020
Morton et al 2019 Plant J
Liu et al 2019 BMC PB on Populus (another tree species)

Line 85 : dtat on Salt and Casuarina equisetifolia are not presented. There are papers
(Ngom et al, 2016 FIPS; Djighaly et al, 2018 annals of forest science). There are data about casuarina in salted condition and authors should see if their observation are similar or not.
Line 95 : Why the fact to have genome gives opportunity? You should develop.
Line 98: Root = first perception  give a reference
Line 100 : explain why. I do not exactly agree with that. Roots are sometimes difficult to work with ; RNA extraction can be tricky with polyphenol and polysaccharides, presence of substrat … At the tissue level, structure is not simple …
Line 104 : cell structure is absolutely not connected with your transcriptomic data. It’s a pity. You should try to see if some DEG are linked to membrane integrity, or to a stage of cells structure degradation.

Line 109 : OK: Can you explain how and which one you will use? This part is absolutely not discussed.

Material and method
Line 120 : why did you choose this time course (previous results ? Please explain. Line 121-122: unclear. Please rewrite.
L1329-132: be sure that these statistics test are adapted to your experiments, generally with a low number of replicates (3), you should use non-parametric tests. Get help from a statistician.

Line 133 : title not adapted. You do histology at two level : tissue level and ultrastructure…but not only electron microscopy
Line 145: Define ‘new root’? How do you select them? Which time course. The sample names should be here. How many replicates …
Line 151 : 20mg seems a bit overestimated for total RNA.. Please verify.
Line 152 precise which Illumina sequencing; Size of the reads?

As a matter of clarification, I would group : RNA extraction for illumina with the mRNA seq experiments parts.
Moreover, I would group RNA extraction for QPCR with QPCR.
Line 158 : & 159 : 2 replicates only? This not the standard for publication of transcriptomic (generally 3 replicates are needed). This also can be a problem with the results of clustering. You clustered 1H and 6h and 24H was grouped with 168h. I am a bit surprised that 24h of salt stress is close to 168h which makes 7days .?...You should precise. Change also the sample names, as CK is troubling (CK means generally cytokine). I would prefer control.
Line 181-185 : unclear. Please reformulate and clarify.
Line 186: The method used is old fashioned for ortholog finding. You should use other tools (orthofinder …). If you use BLASTN you cannot talk or orthologs. By the way, the reference cited is not adapted; you should find another one describing the methodology you used.

Results
Morphological changes in roots
Line 202 : did you compare this observation with the one of Ngom et al 2016 or Djighaly et al 2018?
Line 202 – 205 : not clear with the figure.
Line 206 : Pictures are too small to see the description you do.
Line 214: blocked the deleterious ion transport = you cannot say that. With these observation, you have no proof of that.
Fig 1 is cited …but Differences are really not clear for the root. Please explain more clearly

Changes in ion content
You should describe a little more the table and give values.

Ultrastructural changes in roots
Put this part before the change in ion content.
Line 236 : Did you count?

Response of differentially expressed genes to salt stress in C equisetifolia
L292 : dynamic changes : how do you evaluate that ?

Figure 4A :is cited .. but I have a problem with it. See below. Considering Fig 4A : how can you cluster H1 and H6 and the H24 and H168: patterns are really different. Please explain what is TPM. I cannot understand Fig 5. I don’t understand the fig (see below).

This results part should be more organised; in its present form it is confusing and we cannot really see what is the finding of the study regarding the initial question.
Fig S2 is really interesting and should be put as principal figure.

There is a strong part focusing of transcription factors and the paper of Diédhiou et al. BMC Plant Biology 2014 is not cited, although it is a study of FT in actinorhizal species (Casuarina and alnus).
Papers of Graca et al Int J Mol Sci. 2019 should also be considered, same for Scotti-Campos P, J Plant Physiol. 2016

Discussion
Considering all these remarks, paper should be completed and rewritten. Of course discussion should be updated regarding the new results form.
Line 332 : ‘High level of salinity’: can you precise what is a high level? Maybe you can compare to sea water ?
Line 337-340 : sentence is unclear.
Line 341 : Not clear.. Please give more information.
Line 342 : Histological = no visible change…but what about MET?
Line 348 : It could be interesting to compare your resultas to other species tolerant and not tolerant. My concern is : is there a transcriptomic profile linked to salt tolerance ? What makes the Casuarina equisetifolia more tolerant compared to other species?
Part on PCD should be placed after the ROS part.
You should comment on what will be the next step of the study. What will you do with all these DEG? Will you develop functional studies to validate your hypotheses?

FIGURE 1 : legend are not descriptive enough. Normally the legend should be very clear and precise and autonomous regarding the text. Then root pictures are big and I don’t understand why there are two range of root picture; This should be explained. Fig 1B : pictures are too small and we cannot distinguish what is described in the results part. Scale bars should be specified for roots and more visible for 1 B;

TABLE 1 :Legend is not specified enough. Statistics should be mention. How many replicates …

FIGURE 2 : Legend not specified enough. Picture are too small. Put some of them as Sup data and focus on the most important and then you can put them in bigger size.

FIGURE 3 : Incomplete legend.

FIGURE 4 Incomplete legend. How did you obtain Fig 4A: from what I understand, this is just raw level of mapping? It is not based on Fold change. Please explain. Fig 4B: too small. Fig 4C: not clear and should be in a separated figure.

FIGURE 5: legend are not precise; What is this figure; Is it a comparison of DEG and QPCR? Please explain. And what are the to range of histogram (TPM and relative expression). It is really confusing and unclear.

---

## Round 0.2 · Minor Revisions

The original Academic Editor is no longer available and so I have taken over handling the submission.

The manuscript reads well and makes good points; however, the availability of the described data appears lacking. How does one access the described sequence IDs, such as CCG000005? Without a data foundation the reader has no place to begin a path toward validation. The mention that the raw data is in an SRA resource does not provide links to the gene IDs presented within the manuscript. The manuscript is in need of revision and clarity to link the gene IDs to the reader.

---

## Round 0.3 · accepted · Accept

Thank you for adding suggested edits to the manuscript. The manuscript reads well and the DEGs highlighted are available through the SMRT website. In case the website maintenance may come into jeopardy you may want to consider plans to have the website or data located at a national repository. In general the manuscript reads and describes observation clearly. I have noted a few lines which may require editing listed below. I feel the manuscript is ready to move forward and will list it as accepted. Congratulations.

EDITS
LINE NO: / BEFORE / AFTER / [COMMENTS]
LINE 88: / and Frankia Ceql strain / and the Frankia Ceql strain / [.]
LINE 92: / occurrs / occurs / [.]
LINE 106: / genomic data. / genomic data were added. / [.]
LINE 123: / The method refers / The methods used here refer / [.]
LINE 124: / ), rooted cuttings of / ). Rooted cuttings of / [.]
LINE 173: / sam files were converted into bam files / SAM files were converted into BAM (binary) files / [.]
LINE 174: / reads to references sequences / reads to reference sequences / [.]
LINE 193: / modules. / module parameters. / [.]
LINE 195: / Wolfe. / Wolfe (citation?). / [ Add citation.]
LINE 222: / with Ccl3 or CeD, / with Frankia Ccl3 or CeD strains, / [.]